# Gut Microbial Metabolites of Tryptophan Augment Enteroendocrine Cell Differentiation in Human Colonic Organoids: Therapeutic Potential for Dysregulated GLP1 Secretion in Obesity

**DOI:** 10.3390/ijms26157080

**Published:** 2025-07-23

**Authors:** James Hart, Hassan Mansour, Harshal Sawant, Morrison Chicko, Subha Arthur, Jennifer Haynes, Alip Borthakur

**Affiliations:** Department of Biomedical Sciences, Marshall University Joan C. Edwards School of Medicine, Huntington, WV 25755-0001, USA; hart120@marshall.edu (J.H.); mansour3@marshall.edu (H.M.); sawantha@marshall.edu (H.S.); chicko1@marshall.edu (M.C.); arthursu@marshall.edu (S.A.); haynesje@marshall.edu (J.H.)

**Keywords:** enteroendocrine cells, chromogranin A, GLP1, intestinal organoids, tryptophan metabolites

## Abstract

Enteroendocrine cells (EECs) are specialized secretory cells in the gut epithelium that differentiate from intestinal stem cells (ISCs). Mature EECs secrete incretin hormones that stimulate pancreatic insulin secretion and regulate appetite. Decreased EEC numbers and impaired secretion of the incretin glucagon-like peptide-1 (GLP1) have been implicated in obesity-associated metabolic complications. Gut microbial metabolites of dietary tryptophan (TRP) were recently shown to modulate ISC proliferation and differentiation. However, their specific effects on EEC differentiation are not known. We hypothesized that the gut microbial metabolites of dietary tryptophan counteract impaired GLP1 production and function in obesity by stimulating EEC differentiation from ISCs. We utilized complementary models of human and rat intestines to determine the effects of obesity or TRP metabolites on EEC differentiation. EEC differentiation was assessed by the EEC marker chromogranin A (CHGA) levels in the intestinal mucosa of normal versus obese rats. The effects of TRP metabolites on EEC differentiation were determined in human intestinal organoids treated with indole, a primary TRP metabolite, or the culture supernatant of *Lactobacillus acidophilus* grown in TRP media (LA-CS-TRP). Our results showed that the mRNA and protein levels of CHGA, the EEC marker, were significantly decreased (~60%) in the intestinal mucosa of high-fat-diet-induced obese rat intestines. The expression of the transcription factors that direct the ISC differentiation towards the EEC lineage was also decreased in obesity. In human organoids, treatment with indole or LA-CS-TRP significantly increased (more than 2-fold) CHGA levels, which were blocked by the aryl hydrocarbon receptor (AhR) antagonist CH-223191. Thus, the stimulation of EEC differentiation by colonic microbial metabolites highlights a novel therapeutic role of TRP metabolites in obesity and associated metabolic disorders.

## 1. Introduction

Intestinal epithelial enteroendocrine cells (EECs) are specialized transepithelial signal transduction conduits that respond to luminal nutrients by secreting peptide hormones that regulate the digestion, absorption, and metabolism of nutrients [1,2,3,4]. In particular, two incretin hormones, glucagon-like peptide 1 (GLP1) and glucose-dependent insulinotropic peptide (GIP), are secreted by specific EEC subtypes in response to luminal nutrients. GLP1, produced and secreted by L-type EECs predominantly localized in the colon, is transported via portal blood and neurons to various target organs [2,5,6]. For example, GLP1 stimulates insulin secretion by pancreatic β-cells (insulinotropic effect), whereas in the brain, it generates satiety signals, thereby regulating appetite and food intake [4,6,7,8,9]. Thus, altered production and release of GLP1 by L-cells could play a major role in metabolic disorders such as obesity, type-2 diabetes (T2D), and hypertension [10]. Experimental and clinical evidence suggests that impaired postprandial insulin release and dysregulated appetite control in obesity could partly be due to defective GLP1 secretion and action [11,12,13]. Indeed, the metabolic benefits of gastric bypass have been partly attributed to improved GLP1 production by EECs [14,15]. Further, GLP1 receptor agonists have recently been approved as effective obesity therapeutics for weight loss [16,17], indicating that defective EEC function could be a key factor in obesity.

Altered GLP1 in obesity could partly occur due to the impaired differentiation of EECs from intestinal stem cells (ISCs), causing reduced EEC numbers. Specific transcription factors and signaling pathways direct ISC differentiation towards the EEC lineage [18,19]. Differentiation of crypt-base-located ISCs towards the secretory lineage (enteroendocrine, goblet, and Paneth cells) is governed by the transcription factor ATOH1 (MATH1 in rodents). Next, the sequential expression of two other basic helix–loop–helix transcription factors, Neurogenin 3 (Ngn3) and NeuroD1, is required for terminal differentiation to EECs. Recent studies have shown significant downregulation of these transcription factors and a parallel decrease in CHGA-positive EECs in the intestinal mucosa of obese and T2D patients [20,21].

EECs constitute only ~1% of the total intestinal epithelial cells and are dispersed throughout the epithelial layer between enterocytes. The extremely low number of EECs has been a great hurdle in performing mechanistic studies pertaining to EEC regulation. To overcome this, new technologies have emerged over the past decade, enabling an in-depth analysis of EECs [22]. Among the newer advances, intestinal organoids have recently been used to investigate EEC function and represent a promising model for EEC research [23,24]. Organoid cultures have been shown to contain different types of EECs and to retain their regional identity about the profile of gut hormones produced [22,25]. Other studies have shown that EECs generated within intestinal organoids are responsive to a range of physiological stimuli and are secretion-competent [26].

Diet and gut microbial products play a critical role in directing ISC differentiation towards absorptive versus secretory lineage epithelial cells [27]. In this regard, recent studies have shown that microbial metabolites of tryptophan (TRP) maintain ISC homeostasis via activation of aryl hydrocarbon receptor (AhR) [28]. High dietary TRP acts as a stimulator that promotes the growth of TRP-metabolizing bacteria and the generation of AhR ligands, which drive beneficial functions, such as promoting intestinal barrier functions and maintaining ISC homeostasis [29]. A dysregulated AhR pathway has been shown to cause aberrant ISC proliferation by affecting Wnt/*β*-catenin signaling [30]. Gut microbiota that express tryptophanase, such as *Lactobacilli*, metabolize TRP into indoles. Indoles and indole derivatives, such as indole-3-aldehyde, indole-3-acetic acid, and indole-3-propionic acid, are important signaling molecules [31] and potent AhR activators that mediate AhR’s effects on ISCs. Whether the microbial TRP metabolites directly affect EEC differentiation from ISC, however, is not known. In the current study, we first determined the effects of obesity on the expression of the transcription factors (MATH1, NGN3, and NeuroD1) that govern EEC differentiation from ISC, which subsequently affect the generation of chromogranin A (CHGA)-positive EECs. We used high-fat-diet (HFD)-induced obese Sprague Dawley rats to study EEC dysregulation in obesity. Next, we sought to determine the effects of gut bacterial metabolites of TRP on EEC differentiation and to delineate the underlying mechanisms utilizing a model of human colonic organoids. Here, we report for the first time a direct stimulation of EEC differentiation in human colonic organoids by indole and the TRP metabolites present in the culture supernatant of *Lactobacillus acidophilus* (LA) via an AhR-dependent mechanism.

## 2. Results

### 2.1. Enteroendocrine Cell Number Is Decreased in the Obese Rat Intestinal Mucosa

To determine the effect of obesity on EEC differentiation in vivo, we measured the number of EECs in a rat model of diet-induced obesity. Chromogranin A (CHGA), an abundantly expressed protein in EECs, is considered the best marker for broadly identifying EECs. Utilizing a fluorescently labeled CHGA antibody, EECs from control-diet (CD)- or high-fat-diet (HFD)-fed Sprague Dawley rat small intestinal mucosa were sorted and counted by flow cytometry. As shown in Figure 1, the number of CHGA-positive cells decreased in the small intestinal mucosa of HFD compared to CD rats, with a corresponding increase in CHGA-negative cells.

### 2.2. Chromogranin A mRNA and Protein Levels Are Decreased in Obese Rat Intestinal Mucosa

To verify the decrease in the number of EECs in obese rats, we examined the total expression levels of CHGA in the intestinal mucosa. CHGA mRNA and protein levels in the small intestinal and colonic mucosa of CD and HFD rats were measured by qRT-PCR and Western blot, respectively. CHGA-positive cells (EECs) and GLP1-positive cells (L-type EECs) were also assessed by immunofluorescence studies of sections of CD and HFD rat small intestinal mucosa. As shown in Figure 2, CHGA mRNA in small intestinal (A) and colonic (B) mucosa, CHGA protein in small intestinal (C) and colonic (D) mucosa, and immunostaining of CHGA-positive (E) and GLP1-positive (F) cells were significantly decreased in HFD compared to CD rats.

### 2.3. Enteroendocrine Cell Differentiation Transcription Factors Are Decreased in Obese Rat Intestinal Mucosa

Previous studies have shown that the transcription factors required for directing ISC differentiation towards the EEC lineage (MATH1, NGN3, and NeuroD1) were decreased in the intestinal mucosa of obese patients with a corresponding reduction in the number of CHGA-positive EECs [20,21]. In the current study, we measured the mRNA levels of these transcription factors in the small intestinal mucosa of HFD and CD rats by real-time qRT-PCR. As shown in Figure 3, the mRNA levels of MATH1, NGN3, and NeuroD1 were significantly decreased in the intestinal mucosa of HFD rats compared to CD rats.

### 2.4. Indole, a Gut Microbial Metabolite of Tryptophan, Increases Chromogranin A mRNA and Protein Levels in Human Intestinal Organoids

Gut microbial metabolites are known to modulate EEC differentiation from ISCs [32]. Recent studies have also shown decreased TRP metabolites and TRP-metabolizing gut bacterial species in obesity [33,34]. The supplementation of TRP metabolite or *Lactobacillus* species improved obesity-associated dysregulation of barrier function and GLP1 secretion in the jejunum and colon [33]. Therefore, to determine whether TRP metabolites regulate EEC differentiation in human intestinal cells, we examined the effects of indole, a gut bacterial metabolite of dietary TRP, in human intestinal organoids. Organoids generated from colon biopsies of normal human subjects were treated with indole (0.05–2.0 mM) for 24 h. EEC differentiation was assessed by measuring mRNA and protein levels of CHGA. The colon organoids were chosen for two reasons: (1) bacterial metabolites, including TRP metabolites, are predominantly produced by the fermentation of undigested dietary ingredients in the colon; (2) GLP1-producing L-type EECs are most abundant in colonic epithelium. The results of this experiment (Figure 4) showed that indole treatment dose-dependently increased CHGA mRNA levels (A). Consistent with increased mRNA expression, indole treatment (200 µM) significantly increased CHGA protein levels (B).

### 2.5. Indole Effects on Enteroendocrine Cell Differentiation Are Mediated via Aryl Hydrocarbon Receptor Activation

Aryl hydrocarbon receptor (AhR), a ligand-activated nuclear receptor and transcription factor, has recently been implicated in mediating host–microbiota crosstalk [35]. Since the TRP metabolites are potent agonist ligands of AhR, we determined whether indole’s effects on regulating EEC differentiation are mediated via AhR activation. Organoids were treated with indole (200 uM) in the presence or absence of a selective AhR antagonist CH-223191 (1 µM), and mRNA and protein levels of CHGA were measured. AhR pathway activation was assessed by measuring the induction (mRNA level) of its primary target gene Cyp1A1. As shown in Figure 5, indole treatment increased Cyp1A1 mRNA (A), indicating AhR activation, which caused a significant increase in CHGA mRNA (B) and protein (C) levels. The AhR antagonist (CH223191) alone did not significantly alter CHGA mRNA or protein levels, but it significantly attenuated the indole-mediated increase in CHGA mRNA (B) and protein (C) levels.

### 2.6. Culture Supernatant of Lactobacillus Acidophilus (LA) Grown in Tryptophan Medium Increases Chromogranin A mRNA in Human Colonic Organoids

Several bacterial metabolites have been shown to affect ISC differentiation towards secretory versus absorptive cell lineages of the intestinal epithelial layer [27]. To assess the effect of TRP metabolites directly obtained from the bacterial growth media on EEC differentiation, LA, a known TRP-metabolizing species, was grown overnight in Mann–Rogosa–Sharpe (MRS) media with or without 2 mM TRP. A bacteria-free culture supernatant (CS) of LA was then used to treat human intestinal organoids for 24 h. To determine the dose dependence, the CS was diluted in the organoid culture media at ratios of 1:2, 1:10, 1:20, 1:50, and 1:100. There was no effect on CHGA levels when organoids were treated with 1:50 and 1:100 diluted CS. Based on this dose response, we chose a 1:10 dilution of CS to treat the organoids for 24 h. As shown in Figure 6, both types of LA-CS (obtained from LA overnight culture with TRP or without TRP) significantly increased CHGA mRNA (A). However, this increase was significantly higher with LA-CS containing TRP metabolites (LA grown in the presence of TRP) compared to LA-CS obtained from culture media without TRP. Further, LA-CS-TRP effects, similar to indole effects, on the stimulation of CHGA mRNA (B) and CHGA protein (C) were found to be AhR-activation-dependent. These results suggest that TRP metabolites produced by LA mediate the effects of LA-CS-TRP in stimulating CHGA. The stimulation of CHGA by LA-CS obtained from overnight culture without TRP was presumably due to other metabolites of LA, such as short-chain fatty acids (SCFAs), that were present in the CS.

## 3. Discussion

The systemic impact of EEC-secreted GLP1 on food intake, appetite, and glucose homeostasis and its role in the pathogenesis and treatment of metabolic disorders, including obesity, have been the subject of extensive investigation. However, the exact mechanisms of how GLP1 secretion is affected in obesity are not fully understood and appear to be multifactorial. The levels of GLP1 that are secreted exclusively by L-type EECs of the epithelium could be affected when (i) the L-cell number in the epithelium is decreased, (ii) GLP1 synthesis in L-cells is impaired at the transcriptional, post-transcriptional, or posttranslational level, or (iii) mechanisms that orchestrate GLP1 secretion in response to nutrient sensing by L-cells is affected. One or more of these factors could be responsible for the reduced GLP1 secretion in obesity. Until recently, most of the studies about GLP1 dysregulation in obesity, irrespective of the causative factor (physiological, dietary, or microbial), were focused on the mechanisms of altered production and release of GLP1 by mature EECs [11,36,37,38,39,40,41]. The fact that obese conditions, presumably because of altered dietary patterns and microbial dysbiosis, could also adversely impact EEC differentiation from ISCs, causing a substantial reduction of GLP1-secreting L-cells, has only recently been appreciated [21,42,43,44]. In the current study, we sought to investigate whether altered GLP1 in obesity could, at least partly, be caused by impaired EEC differentiation from ISCs, causing a reduced EEC number.

In a diet-induced rat model of obesity (HFD-induced obesity in Sprague Dawley rats), our results showed decreased mRNA and protein levels of the EEC marker CHGA, suggesting decreased EEC number. Decreased EECs in obesity were also directly demonstrated by sorting the CHGA-positive cells in the mucosa by flow cytometry and by immunofluorescence staining of CHGA-positive cells in mucosal sections of CD versus HFD rats. To specifically demonstrate that GLP1-producing L-type EECs decrease in the obese intestine, we used immunofluorescence studies in mucosal sections of CD and HFD rats with the GLP1 antibody and found that GLP1-positive L-cells also decreased in obesity.

The differentiation of ISCs towards absorptive (enterocytes) versus secretory (enteroendocrine, goblet, and Paneth cells) lineage cells is governed by various transcription factors and signaling pathways. Active Notch signaling stimulates the expression of Hairy/enhancer of split 1 (HES1), a potent repressor for the basic helix–loop–helix transcription factors ATOH1 (also known as MATH1) and neurogenin-3 (NGN3). Whereas the former is important for producing all secretory cells, the latter, along with another critical transcription factor NeuroD1, is the key regulator for EEC cell formation [45]. Mice deficient for NGN3 completely lack all EEC subtypes in the small and large intestines. Conversely, transgenic overexpression of NGN3 greatly increases the generation of all EEC lineages [45]. Our results in HFD-induced obese rats showed a significant decrease in the mucosal expression of MATH1, NGN3, and NeuroD1 compared to the respective control animals. Concomitantly, mRNA and protein levels of the EEC marker CHGA were decreased in HFD rats, suggesting an inhibitory effect of an HFD on EEC differentiation. These results support that altered GLP1 secretion and function in obesity could be partly due to decreased EEC differentiation that results in a decreased GLP1-producing L-cell number.

Currently, the most effective long-term treatment of obesity is bariatric surgery [46], which, however, is not economically viable at the population level because of its high cost [47,48]. A key mechanism of surgery-induced weight loss and diabetes remission is the increased delivery of nutrients to L-type EECs, resulting in highly elevated postprandial levels of GLP-1 that lead to improved glycemia and weight loss [14,49]. To combat dysregulated GLP1 secretion in obesity and type-2 diabetes (T2D), GLP1 mimetics have subsequently been developed as GLP1 receptor agonists [16,50]. Despite this, directly targeting L-cells with agents (nutritional and/or gut microbial) to stimulate the endogenous production and secretion of GLP1 is still highly relevant as an alternative or combined treatment of obesity for various reasons [51]. The nutritional or microbial approach has minimal or no potential adverse side effects, including GI discomfort, nausea, vomiting, and diarrhea, sometimes observed when using synthetic GLP1 receptor agonists [52,53]. The vagal/neural dependency of part of the GLP1 action (for example, in appetite regulation) might not be accessible for GLP1R agonists [54,55]. Finally, nutritional/microbial agents might lead to the increased secretion of colocalizing hormones, including PYY, GIP, and oxyntomodulin, providing an additive effect on appetite suppression and blood glucose regulation [56].

Diet, gut microbiota, and microbial metabolites, important factors regulating obesity pathogenesis, have recently been shown to play a critical role in directing ISC differentiation towards absorptive versus secretory lineage epithelial cells [32,57,58], thereby playing key roles in the dynamic balance of intestinal epithelial cell composition. For example, an HFD has been shown to cause a transcriptional shift in cell lineage trajectory from secretory to absorptive [59]. There is a substantial decrease in all secretory lineages and an increase in enterocytes over 7 days of an HFD in the proximal intestine [59]. However, the studies investigating the effects of microbiota and metabolites on ISC differentiation are relatively limited. Short-chain fatty acids, gut microbial metabolites of dietary fiber, have been shown to regulate ISC proliferation and stimulate goblet cell differentiation [27]. A recent study in pre-clinical and clinical settings showed that metabolic syndrome is associated with a reduced abundance of TRP-metabolizing bacteria in the gut microbiota [33]. Therefore, the second part of the current study was aimed at determining the effects of indole, a key TRP metabolite, on EEC differentiation, showing increased EEC differentiation (which logically increases GLP1-producing L-cell number also) in human colonic organoids. To provide the effects of the full profile of TRP metabolites present in the culture supernatant of LA, a predominant TRP-metabolizing probiotic bacteria, we used supernatants obtained from an overnight culture of LA in (1) MRS media alone (designated as LA-CS) and (2) MRS media containing 2 mM L-tryptophan (designated as LA-CS-TRP). Both LA-CS and LA-CS-TRP stimulated EEC differentiation as compared to MRS alone. However, LA-CS-TRP stimulation was significantly higher compared to LA-CS stimulation. LA-CS stimulation was presumably due to other metabolites of LA, such as SCFA, lactate, and maybe even some tryptophan metabolites. When LA grew with 2 mM TRP, the concentration of the TRP metabolites in the CS increased; therefore, the additional increase in EEC differentiation by LA-TRP appears to be due to increased TRP metabolites generated by LA from the added substrate (TRP). We named it broadly as TRP metabolites of LA. Our novel findings showing the stimulation of EEC differentiation by gut microbial TRP metabolites highlight the promising potential of targeting L-cells to develop obesity therapeutics. Intestinal organoids have emerged as a valuable tool to study the differentiation, function, and regulation of EECs. Organoid cultures have been shown to contain different types of EECs and to retain their regional identity regarding the profile of gut hormones produced [22,25]. Other studies have shown that EECs generated within intestinal organoids are responsive to a range of physiological stimuli and are secretion-competent [26].

Mechanistically, the TRP metabolite-mediated stimulation of EEC differentiation in human intestinal organoids observed in our studies showed the role of aryl hydrocarbon receptor (AhR). This ubiquitous receptor, initially identified as a conserved sensor of small xenobiotic molecules [60], was later characterized as a transcription factor activated by several endogenous and exogenous ligands, including indole-related molecules produced by the microbiota. Indeed, AhR is now considered a pivotal mediator of gut microbiota–host epithelium crosstalk [35]. Recently, increasing attention has been focused on AhR due to its role in the regulation of inflammation, immunity, intestinal barrier function, and intestinal microecology [61]. Importantly, a recent study has demonstrated that metabolic syndrome is associated with the reduced capacity of microbiota to produce TRP metabolites known to activate AhR. Indeed, supplementation with indole, an AhR agonist, or *Lactobacillus reuteri*, a strain with a high AhR ligand-production capacity, led to the improvement of metabolic impairments, such as glucose dysmetabolism, via improving intestinal barrier function and GLP1 production [33]. Since in our studies, indole or LA-derived TRP metabolites did not alter the expression of the EEC differentiation transcription factors (Hes 1, NGN3, and NeuroD1) (results not shown), the AhR activation by these metabolites presumably affected Wnt/Notch signaling to stimulate EEC differentiation. Previous studies have reported the AhR-mediated modulation of ISC differentiation and stem cell fate via its effects on Wnt/Notch signaling [28,62].

A major limitation of our current study is that the effects of TRP metabolites on EEC differentiation were determined in colonic organoids generated from biopsies of normal human subjects. The molecular players and signaling pathways involved in the regulation of EEC differentiation could be different in the normal and obese conditions. Studying the differential mechanisms of EEC regulation and the effects of TRP metabolites utilizing intestinal organoids derived from healthy subjects versus obese patients will be paramount to defining novel pathways to target for the development of effective obesity therapeutics.

## 4. Materials and Methods

### 4.1. Antibodies and Chemicals

The following antibodies were used for this study: CoraLite^®^ Plus 488-conjugated chromogranin A polyclonal antibody (Proteintech, Rosemont, IL, USA, catalog# CL488-10529) for flow cytometry, unconjugated polyclonal chromogranin A antibody (Novus Biologicals, Centennial, CO, USA, catalog # NB120-15160) for Western blot and immunofluorescence studies, and monoclonal β-actin antibody (Santa Cruz Biotechnology, Santa Cruz, CA, USA, Catalog #sc47778) as the loading control. Indole was purchased from Sigma Aldrich, and CH-332101 was procured from Tocris.

### 4.2. Rat Model of Obesity

For the diet-induced obesity model, Sprague Dawley rats (Charles River Laboratories, Wilmington, MA, USA) were fed a control diet (CD, 10% fat, D12450) or high-fat diet (HFD, 60% fat, D12492) procured from Research Diets (New Brunswick, NJ, USA) for 18 weeks. The rats were then sacrificed, and the intestinal mucosa was scraped and processed for subsequent experiments. The animal studies were approved by the Institutional Animal Care and Use Committee (IACUC) of Marshall University (protocol #756).

### 4.3. Human Colonic Organoids

Two lines of frozen human colonic organoids (described in Table 1) were obtained from TCM DDC GEMS 3D Organoids Core, Dr. Mary Estes Lab, Department of Molecular Virology and Microbiology, Baylor College of Medicine (BCM), Houston, TX, USA, and stored in liquid nitrogen until used. Before experiments, the frozen colonoid lines were thawed and expanded as 3D organoid cultures grown in 50 µL Matrigel domes and overlaid with 650 µL of a colonoid culture media prepared by BCM DDC Core (WRNE-Nico: Wnt 3a, R-spondin, Noggin, EGF, without nicotinamide) [63] + 10 µM Y27632 + 2.5 µM and cultured in a 37 °C CO_2_ incubator. Organoids were passaged after 6–7 days and were used for experiments on day 5 after passage. On this day, organoids were treated for 24 h with indole (0.1–2 mM) or a bacteria-free conditioned culture supernatant of *Lactobacillus acidophilus* grown overnight in the presence or absence of 2 mM TRP. Indoles in the indicated concentrations were prepared, and LA culture supernatants were diluted 1:10 in differentiation media (WRNE medium without L-WRN conditioned medium, nicotinamide, and SB202190; addition of 5% Noggin conditioned medium).

### 4.4. Bacterial Culture and Preparation of Conditioned Culture Supernatant

*Lactobacillus acidophilus* (strain # 4356) obtained from American Type Culture Collection (ATCC, Manassas, VA, USA) was grown in Mann–Rogosa–Sharpe (MRS) broth (Difco Laboratories, Detroit, MI, USA) with or without 2 mM TRP for 24 h at 37 °C without shaking, as previously described by us [64]. The cultures were then centrifuged at 3000× *g* × 10 min at 4 °C. The supernatant, filtered through a 0.22 μm filter (Millex, Millipore, Billerica, MA, USA) to sterilize and remove all bacterial cells, was designated as the conditioned culture supernatant (CS).

### 4.5. Sorting of Chromogranin A-Positive EECs from Rat Intestine by Flow Cytometry

We used flow cytometry to separate the EECs of rat intestinal mucosa as per a previously described method [65], with appropriate modifications required to label intracellular chromogranin A in EECs. Briefly, the small intestine of the rat was cut longitudinally, washed in cold PBS, and incubated in 2% (vol/vol) fetal calf serum in RPMI 1640 (Gibco) containing 0.5 mM EDTA for 15 min at 37 °C. Cells were filtered with a 70 μm cell strainer (Becton Dickinson, Franklin Lakes, NJ, USA) and blocked with Fc Block (2.4G2; Becton Dickinson) for 5 min at room temperature. To enable the fluorescent antibody labeling of intracellular chromogranin A in EECs, cells were fixed with a 1% paraformaldehyde–PBS solution for 20 min and subsequently permeabilized with 0.1% saponin in 2% BSA and 0.2% RNaseOUT in PBS for 15 min. The primary antibody CoraLite^®^ Plus 488-conjugated chromogranin A polyclonal antibody (Proteintech, catalog# CL488-10529) was added and incubated for 60 min. Dead cells were excluded by propidium iodide (PI) staining. During sorting experiments, cells were placed in FACS buffer with an RNase inhibitor (Life Technologies, Carlsbad, CA, USA) and analyzed by Agilent Novocyte 3000 Flow Cytometer. Data analysis was performed using NovoExpress Software vX (Agilent, Santa Clara, CA, USA).

### 4.6. Total RNA Isolation and Real-Time Quantitative Polymerase Chain Reaction

RNA was extracted from rat intestinal mucosal samples and human intestinal organoids using an RNeasy mini kit (Qiagen, Germantown, MD, USA) following the manufacturer’s protocol. For organoid cultures, organoid-containing Matrigel domes were washed three times with cold PBS, gently scraped in ice-cold Cell Recovery Solution (Corning), and transferred to prechilled 15 mL centrifuge tubes. Samples were incubated on ice for 45 min, and tubes were gently rocked every 15 min. Released cells were then collected by centrifugation at 200× *g* for 5 min (4 °C), washed with ice-cold PBS, and lysed by adding RLT lysis buffer provided in the kit before total RNA extraction. RNA was reverse-transcribed and amplified using a Brilliant SYBR Green QRT-PCR Master Mix kit (Agilent Technology, Santa Clara, CA, USA). Human or rat chromogranin A was amplified using species- and gene-specific primers custom-designed by Thermo Fisher Scientific (Hillsboro, OR, USA) with GAPDH or β2-microglobulin (for organoids) amplified as the internal controls. The sequences of the primers used in this study are given in Table 2.

### 4.7. Western Blotting

Tissue lysates from scraped rat intestinal mucosa or the organoids were prepared using RIPA buffer (Thermo Fisher Scientific). Before adding RIPA buffer, cells from organoid cultures were prepared as described above for RNA extraction. The samples were run on 10% SDS-PAGE and then transferred onto PVDF membranes. Immunoblotting was carried out with anti-CHGA antibody (dilution 1:1000) and β-actin antibody (dilution 1:2000) as the loading control. Bands were visualized with enhanced chemiluminescence detection reagents. Relative band intensities were measured utilizing Image Lab software, version 6.1 (BioRad, Hercules, CA, USA).

### 4.8. Immunostaining

Sections of small intestinal tissues from CD and HFD rats were snap-frozen in optimal cutting temperature embedding medium. For immunostaining, 5 μm frozen sections were fixed with 1% paraformaldehyde in PBS for 10 min at room temperature. Fixed sections were washed in PBS, permeabilized with 5% Nonidet P-40 for 5 min, and blocked with 5% normal goat serum (NGS) for 30 min. Tissues were incubated with Chromogranin A antibody (1:100) in PBS with 1% NGS for 90 min at room temperature. After being washed, sections were incubated with Alexa Fluor 594-conjugated goat anti-rabbit IgG and Alexa Fluor 488-conjugated phalloidin (5 U/mL; Invitrogen) for 60 min. Sections were then washed and mounted under cover slips using Slowfade Gold antifade with DAPI reagent (Invitrogen). Sections were imaged using a Carl Zeiss LSM 510 laser-scanning confocal microscope equipped with ×20 water immersion objective.

### 4.9. Data Analysis and Statistics

Excel version 16.85 (Microsoft, Redmond, WA, USA), Image Lab 5.2.1 (BioRad Laboratories, Hercules, CA, USA), and Prism 8.0 (GraphPad Software, San Diego, CA, USA) software were used for data acquisition, analysis, and presentation. The tests used for the data analysis in Prism were unpaired *t*-tests and one-way and multifactorial Analysis of Variance (ANOVA) depending on the data sets, as appropriate. Error bars indicated ± Standard Error of Mean (SEM). Probability values of *p* < 0.05 were considered statistically significant. The Shapiro–Wilk test in Prism was used to confirm a normal distribution of the data sets.

## Figures and Tables

**Figure 1 ijms-26-07080-f001:**
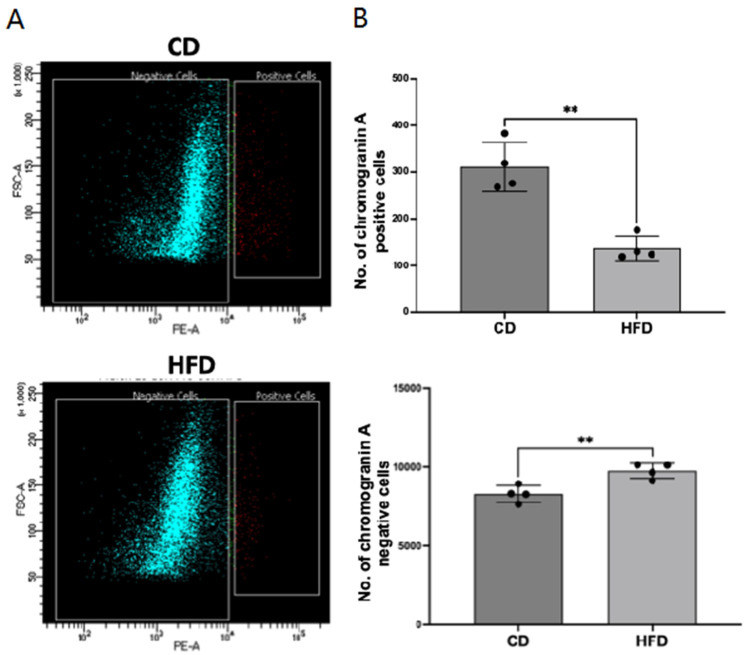
Enteroendocrine cell (EEC) number is decreased in obese rat intestinal mucosa. (**A**). Epithelial cells from the small intestinal mucosa of rats fed a control diet (CD) or high-fat diet (HFD) were sorted for chromogranin A (CHGA)-positive cells (red) by FACS utilizing a fluorescently labeled CHGA antibody, as described in the Methods. (**B**). Comparison of the relative numbers of CHGA-positive versus CHGA-negative (green) epithelial cells in the intestinal mucosa of CD versus HFD rats. Data represent Mean ± SEM. ** *p* < 0.05.

**Figure 2 ijms-26-07080-f002:**
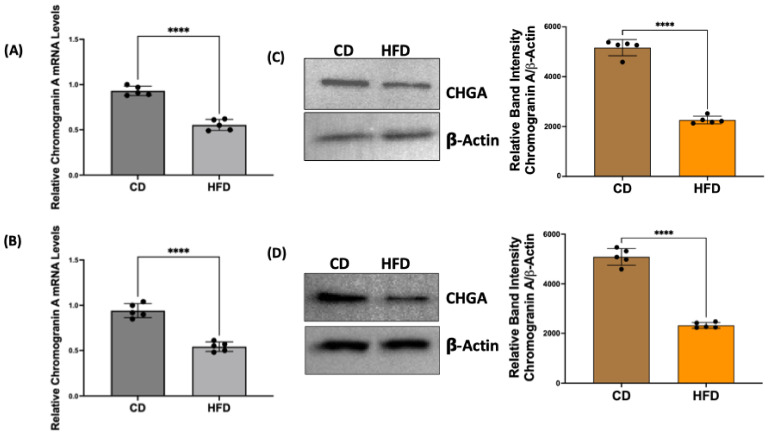
CHGA mRNA and protein levels are decreased in obese rat intestinal mucosa. Total RNA isolated from scraped small intestinal (**A**) and colonic (**B**) mucosa of CD/HFD rats was amplified with rat-specific gene primers for real-time PCR quantification. Data represent the relative expression of CHGA normalized to respective GAPDH mRNA (internal control). Tissue lysates prepared from scraped small intestinal and colonic mucosa of CD/HFD rats were subjected to Western blot analysis using an anti-CHGA antibody. The left panels show representative blots showing the expression of CHGA in CD versus HFD rat small intestinal (**C**) and colonic (**D**) mucosa with β-actin as the loading control. The right panels show the results of a densitometric analysis of the band intensities of CHGA normalized to that of β-actin. Frozen sections of small intestinal mucosa of CD versus HFD rats were immunostained for CHGA (**E**) or GLP1 (**F**). [CHGA (green); GLP1 (red) and nuclei (blue)]. Arrows indicate CHGA-positive (**E**) or GLP1-positive (**F**) cells. Representative images are shown in (**E**,**F**) with scale bar 100 μM. In all figures (**A**–**E**), data represent Mean ± SEM. ** *p* < 0.05; **** *p* < 0.0001.

**Figure 3 ijms-26-07080-f003:**
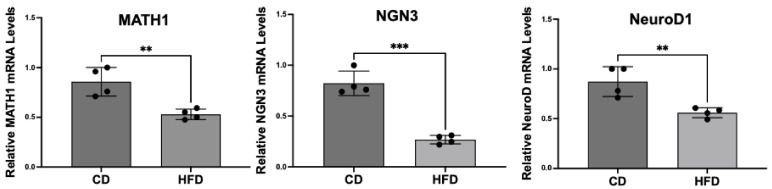
Transcription factors governing EEC differentiation are decreased in the obese rat intestine. Total RNA isolated from scraped small intestinal mucosa of CD and HFD rats was amplified with rat-specific gene primers for real-time PCR quantification. Data represent the relative expression of MATH1, NGN3, and NeuroD1 normalized to respective GAPDH mRNA (internal control). Data represent Mean ± SEM. ** *p* < 0.05; *** *p* < 0.001.

**Figure 4 ijms-26-07080-f004:**
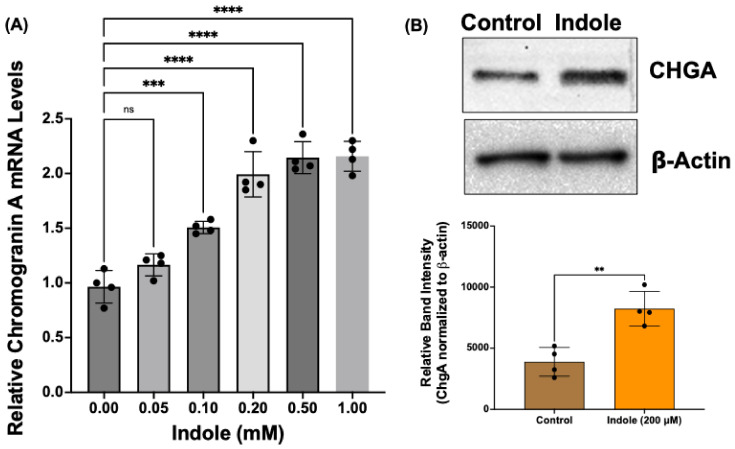
Indole, a gut microbial metabolite of tryptophan (TRP), increases CHGA mRNA and protein levels in human colonic organoids. (**A**) Total RNA isolated from control and indole-treated colon organoids was amplified with human-specific gene primers for real-time PCR quantification. Data represent the relative expression of CHGA normalized to respective β2-microglobulin (B2M) mRNA (internal control). (**B**) Lysates prepared from control and indole-treated (200 μM) organoids were subjected to Western blot analysis. The upper panel is a representative blot showing the expression of CHGA with β-actin as the loading control. The lower panel shows the results of a densitometric analysis of band intensities of CHGA normalized to that of β-actin. ** *p* < 0.05, *** *p* < 0.001, **** *p* < 0.0001 between groups as indicated.

**Figure 5 ijms-26-07080-f005:**
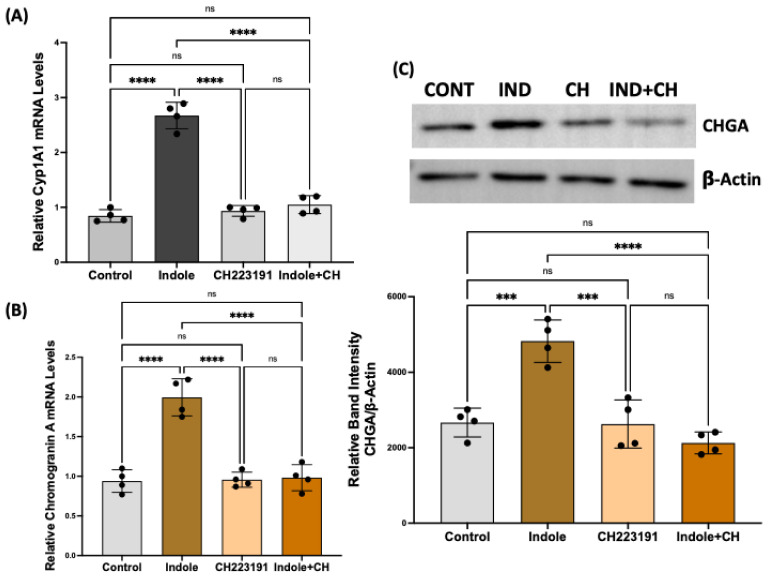
Indole effects on EEC differentiation are mediated via aryl hydrocarbon receptor activation. Total RNA isolated from control and indole (0.2 mM) ± CH223191-treated colonic organoids was amplified with human-specific gene primers of Cyp1A1 (**A**) and CHGA (**B**) for real-time PCR quantification. Data represent the relative expression of CHGA normalized to respective β2-microglobulin (B2M) mRNA (internal control). **** *p* < 0.0001 between groups as indicated. (**C**). Lysates prepared from control and indole (0.2 mM) ± CH223191-treated organoids were subjected to Western blot analysis. The upper panel is a representative blot showing the relative expression of CHGA with β-actin as the loading control. The lower panel shows the results of a densitometric analysis of band intensities of CHGA normalized to that of β-actin. *** *p* < 0.001, **** *p* < 0.0001 between groups as indicated.

**Figure 6 ijms-26-07080-f006:**
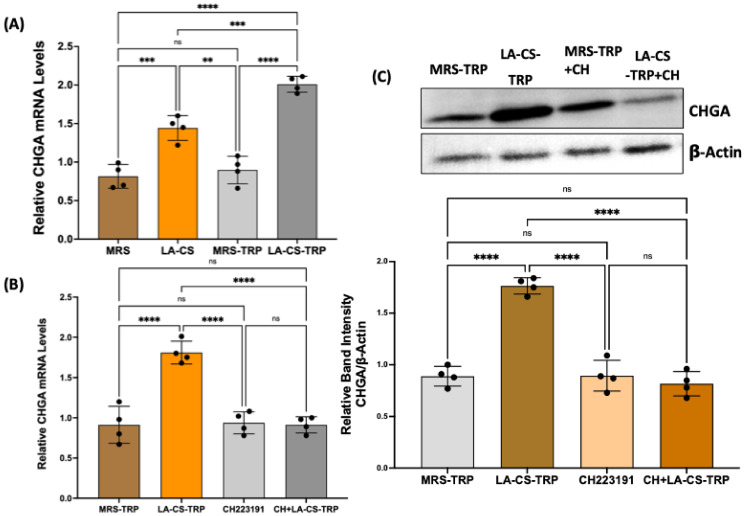
TRP metabolites in *Lactobacillus acidophilus* culture supernatant (LA-CS) stimulate EEC differentiation via an AhR activation-dependent mechanism. (**A**) Total RNA isolated from different intestinal organoid groups (treated with MRS media, LA-CS, MRS-TRP media, and LA-CS-TRP) was amplified with human-specific gene primers (CHGA) for real-time PCR quantification. Data represent the relative expression of CHGA normalized to respective β2-microglobulin (B2M) mRNA (internal control). ** *p* < 0.05, *** *p* < 0.001, **** *p* < 0.0001 between groups as indicated. (**B**) Total RNA samples prepared from control and LA-CS-TRP ± CH223191-treated organoids were subjected to real-time qRT-PCR. Data represent the relative expression of CHGA mRNA normalized to respective β2-microglobulin (B2M) mRNA (internal control). **** *p* < 0.0001 between groups as indicated. (**C**) Lysates prepared from control and LA-CS-TRP ± CH223191-treated organoids were subjected to Western blot analysis. The upper panel is a representative blot showing the relative expression of CHGA in different groups, with β-actin as the loading control. The lower panel shows the results of a densitometric analysis of band intensities of CHGA normalized to that of β-actin. **** *p* < 0.0001 between groups as indicated.

**Table 1 ijms-26-07080-t001:** Description of the human subjects from whom organoids were generated at BCM.

Organoid Line	Origin	Gender	Age	Normal/Diseased
C04	Ascending colon	M	50	Normal
C103	Ascending colon	F	24	Normal

**Table 2 ijms-26-07080-t002:** Gene-specific primers used for real-time PCR analysis of mRNA levels (F: forward primer; R: reverse primer).

Gene	Primer Sequence (5′-3′)
Human *CHGA*	F: CCCTGTGAACAGCCCTATGA R: GGTCTTGGAGCTCCTTCAGT
Human *Cyp1A1*	F: TGGAGACCTTCCGACACTCT R: ACAAAGACACAACGCCCCTT
Human *b2-microglobulin*	F: CTCCGTGGCCTTAGCTGTG R: TTTGGAGTACGCTGGATAGCC
Rat *CHGA*	F: GCATGGGATTCCACAGACCA R: GTGGGGACTTCTTTAGGCCC
Rat *MATH1*	F: CCTAACAGCGATGATGGCAC R: GTCTTCCTAACTGGCCTCGT
Rat *Neurogenin 3*	F: GCGTGGAGTGACCTCTAAGT R: AAAGGGTTGCTGGGTCTCTT
Rat *NeuroD1*	F: CTAACTGATTGCACCAGCCC R: CAAACTCGGTGGATGGTTCG
Rat *GAPDH*	F: TGCACCACCAACTGCTTAGC R: GGCATGGACTGTGGTCATGAG

## Data Availability

The authors confirm that the data supporting the findings of this study are available within the article.

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
