# Peer review of "Gut Microbial Metabolites of Tryptophan Augment Enteroendocrine Cell Differentiation in Human Colonic Organoids: Therapeutic Potential for Dysregulated GLP1 Secretion in Obesity"

_ijms, 2025, doi:10.3390/ijms26157080_

Round 1
Reviewer 1 Report
Comments and Suggestions for Authors
This study investigates an important and timely topic: the role of gut microbial tryptophan metabolites in promoting enteroendocrine cell (EEC) differentiation via AhR activation, with therapeutic implications for obesity-associated GLP1 deficiency. The hypothesis is well-justified, and the use of complementary rat and human organoid models strengthens the work. While the findings are novel and mechanistically insightful, several methodological and interpretive issues require clarification.
Major Concerns:
1. Specificity of L. acidophilus Metabolites:
The study attributes effects of LA-CS-TRP to "TRP metabolites" but does not characterize the metabolomic profile of the supernatant. L. acidophilus primarily produces indole (as noted in the Introduction), but other metabolites (e.g., SCFAs, lactate) or bacterial components (e.g., peptidoglycan) in the supernatant could contribute to AhR activation or EEC differentiation.
Recommendation: Perform LC-MS/MS to quantify indole/IAA/IPA/IAld in LA-CS-TRP vs. control supernatant. Test purified indole derivatives alongside LA-CS-TRP to confirm shared mechanisms.
2. Physiological Relevance of Doses:
Indole was tested at 0.05–2.0 mM, and LA-CS was diluted 1:10. The author should detect the concentration of indole in the LA-CS-TRP supernatant, and set the same concentration for verification.
3. Incomplete EEC Differentiation Analysis:
CHGA is a pan-EEC marker. The study does not quantify GLP1+ L-cells specifically in organoids (only in rat mucosa). Increased CHGA could reflect rises in non-GLP1 EECs (e.g., serotonin-producing cells).
Recommendation: Measure GLP1 gene/protein in organoids after indole/LA-CS-TRP treatment. Co-stain for CHGA and GLP1 to confirm L-cell expansion.
4.Unclear Contribution of AhR to L. acidophilus Effects:
LA-CS-TRP increased CHGA mRNA/protein, and CH-223191 blocked this effect (Fig. 6B–C). However, LA-CS without TRP also increased CHGA (Fig. 6A), suggesting AhR-independent mechanisms.
Recommendation: Test if AhR antagonist blocks LA-CS (without TRP) effects. Acknowledge that non-TRP metabolites in LA-CS may influence EEC differentiation.
Minor Concerns:
- Fig1/2: Clarify if "small intestinal mucosa" refers to duodenum/jejunum/ileum. Regional heterogeneity in EEC density is well-established.
- Organoid Media: The "differentiation media" (Section 4.3) omits key components (e.g., Wnt, R-spondin). Specify how differentiation was induced.
Recommendation:
Major Revision. The manuscript requires additional experiments (metabolomics of LA-CS, GLP1 measurements in organoids) and contextual adjustments (dose justification, obesity models) to fully support its conclusions.
Additional Notes for Authors:
1.The discussion section should integrate the current state of research to elucidate the impact of obesity or a HFD on intestinal LA, and further elaborate on the metabolic pathways through which LA processes tryptophan, thereby enhancing readers' comprehension.
2. Discuss whether AhR activation directly upregulates EEC transcription factors (e.g., NGN3) or acts indirectly via ISC niche modulation.
Author Response
Authors’ Answers to Reviewer 1 Comments
Major Concerns:
Q1. Specificity of L. acidophilus Metabolites:
The study attributes effects of LA-CS-TRP to "TRP metabolites" but does not characterize the metabolomic profile of the supernatant. L. acidophilus primarily produces indole (as noted in the Introduction), but other metabolites (e.g., SCFAs, lactate) or bacterial components (e.g., peptidoglycan) in the supernatant could contribute to AhR activation or EEC differentiation.
Recommendation: Perform LC-MS/MS to quantify indole/IAA/IPA/IAld in LA-CS-TRP vs. control supernatant. Test purified indole derivatives alongside LA-CS-TRP to confirm shared mechanisms.
Answers: Thanks for the useful suggestions. The LA-CS-TRP experiment was intended to provide the effect of all TRP metabolites on EEC differentiation in a more realistic setting that includes the full profile of LA supernatant. In Figure 6, we compared the effects of LA supernatants obtained from overnight culture of LA in (1) MRS media alone (designated as LA-CS) and (2) MRS media containing 2 mM L-Tryptophan (designated as LA-CS-TRP). Both LA-CS and LA-CS-TRP stimulated EEC differentiation as compared to MRS alone. However, LA-CS-TRP stimulation was significantly higher compared to LA-CS stimulation. LA-CS stimulation is presumably due to other metabolites of LA, such as SCFA, lactate, and maybe even some tryptophan metabolites. When LA grew with 2 mM TRP, the concentration of the TRP metabolites in the CS increased; therefore, the additional increase in EEC differentiation by LA-TRP appears to be due to the TRP metabolites generated from the added 2 mM TRP. We named it broadly as the TRP metabolites of LA. Therefore, the characterization and quantification of all the TRP metabolites (indole/IAA/IPA/IAld) is beyond the scope of the studies included in this manuscript. However, in the revised manuscript, we have now better discussed this experiment in the Discussion section (lines 316-331).
Q2. Physiological Relevance of Doses:
Indole was tested at 0.05–2.0 mM, and LA-CS was diluted 1:10. The author should detect the concentration of indole in the LA-CS-TRP supernatant and set the same concentration for verification.
Answers: The physiological concentration of indole in the gut lumen exhibit a broad range (0.25 mM – 2.6 mM) (PMID: 34966278) with significant individual variability, which is also largely dependent on the type of diet consumed and gut microbial composition. Because of this broad physiological range, we determined the dose response of indole on EEC differentiation. Likewise, we did a dose response of LA-CS from 1:2 to 1:100 dilution, and based on that, 1:10 dilution was chosen. We now mention this in the text in the Results section (lines 218-222). Since LA-CS-TRP contains TRP metabolites other than indole (IAA/IPA/IAld), we did not see any rationale for measuring indole concentration in LA-CS-TRP for this manuscript to compare with the experiments using pure indole.
Q3. Incomplete EEC Differentiation Analysis:
CHGA is a pan-EEC marker. The study does not quantify GLP1+ L-cells specifically in organoids (only in rat mucosa). Increased CHGA could reflect rises in non-GLP1 EECs (e.g., serotonin-producing cells).
Recommendation: Measure GLP1 gene/protein in organoids after indole/LA-CS-TRP treatment. Co-stain for CHGA and GLP1 to confirm L-cell expansion.
Answers: As the title of the manuscript says, the aim of this study was to examine the effects of gut microbial TRP metabolites on EEC differentiation, not specifically on L-cell differentiation. Intestinal stem cells (ISCs) may be differentiated towards secretory and absorptive cell lineages, which is well balanced in a healthy state. Studies have reported that in obesity, differentiation of secretory cells, including EECs, decreases, whereas there is increased differentiation towards absorptive lineage cells. Gut microbial metabolites are known to affect ISC differentiation towards secretory versus absorptive lineages. Therefore, this study aimed to determine whether TRP metabolites augment EEC cell differentiation. That’s why we measured CHGA-positive cells in rats fed CD versus HFD and the effects of indole on CHGA-positive cells in human organoids.
Q4. Unclear Contribution of AhR to L. acidophilus Effects:
LA-CS-TRP increased CHGA mRNA/protein, and CH-223191 blocked this effect (Fig. 6B–C). However, LA-CS without TRP also increased CHGA (Fig. 6A), suggesting AhR-independent mechanisms.
Recommendation: Test if AhR antagonist blocks LA-CS (without TRP) effects. Acknowledge that non-TRP metabolites in LA-CS may influence EEC differentiation.
Answers: This concern has already been addressed above, please see A1.
Minor Concerns:
Q1. Fig1/2: Clarify if "small intestinal mucosa" refers to duodenum/jejunum/ileum. Regional heterogeneity in EEC density is well-established.
Answers: Unless otherwise stated, throughout the manuscript, small intestinal mucosa refers to the mucosal scrapings of the entire small intestine (duodenum/jejunum/ileum), and colonic mucosa refers to the mucosal scrapings of the proximal and distal colon.
Q2. Organoid Media: The "differentiation media" (Section 4.3) omits key components (e.g., Wnt, R-spondin). Specify how differentiation was induced.
Answers: These 2 components preserve the stemness of ISCs, promoting proliferation and inhibiting differentiation. They are omitted to augment differentiation of transit amplifying cells to secretory and absorptive lineages, which is governed by specific transcription factors.
Additional Notes for Authors:
Q1. The discussion section should integrate the current state of research to elucidate the impact of obesity or an HFD on intestinal LA and further elaborate on the metabolic pathways through which LA processes tryptophan, thereby enhancing readers' comprehension.
Answers: We have already briefly referred to these aspects in the Introduction. In our opinion, the Discussion section of a research article should be highly focused on discussing the results and should avoid elaborate descriptions of cellular processes (for example, pathways through which LA processes tryptophan, HFD-induced microbial dysbiosis), which have been discussed in detail in a number of review articles. Therefore, we respectfully disagree with the reviewer to elaborate on the suggested aspects that are not directly related to the aims of the current study.
Q2. Discuss whether AhR activation directly upregulates EEC transcription factors (e.g., NGN3) or acts indirectly via ISC niche modulation.
Answers: This has now been included in the discussion section (lines 352-357).
Reviewer 2 Report
Comments and Suggestions for Authors
- IJMS template was not used.
- There are no exact numbers in the abstract's results section.
- List of abbreviation's tryptophan should be written in capital, while all others are so.
- For example, GLP1 stimulates insulin secretion - Please rephrase 'for example'. In the discussion section, as well.
- Next, the sequential expression of two other basic helix-loop-helix transcription factors - Please rephrase 'next'.
- Other studies have shown - Please rephrase.
- The last chapter of the Introduction should be rephrased into an aim, because in this format, this is rather Materials and methods.
- Figure 1: Negative cells title should be positioned a bit upwards. There also seems to be a broken line under HFD.
- Instead of we examined, ... was examined (passive formatting) should be used.
- Figure 2 - The quality of immunofluorescence photos are not the best.
- In the results section, general and literature data should not be included (such as: Aryl hydrocarbon receptor (AhR), a ligand-activated nuclear receptor and transcription
factor, has recently been implicated in mediating host-microbiota crosstalk). - 4.8. Immunosaining - Please correct. - Furthermore, this section contains the detailing of immunofluorescence, which is not the same as immunohistochemistry!
- Data analysis and statistics - Please rephrase to Data and statistical analyses.
Author Response
Authors’ Answers to Reviewer 2 Comments
Q1. The IJMS template was not used.
Answer: The manuscript was written as per the guidelines for authors, and there was no concern from the journal.
Q2. There are no exact numbers in the abstract's results section.
Answer: We have now mentioned fold-change in some of the results.
Q3. List of abbreviations: tryptophan should be written in capital letters, while all others are so.
Answer: Corrected.
Q4. For example, GLP1 stimulates insulin secretion - Please rephrase 'for example'. In the discussion section, as well.
Q5. Next, the sequential expression of two other basic helix-loop-helix transcription factors - Please rephrase 'next'.
Q6. Other studies have shown - Please rephrase.
Q7. The last chapter of the Introduction should be rephrased into an aim, because in this format, this is rather Materials and methods.
Q8. Instead of we examined, ... was examined (passive formatting) should be used.
Q9. Data analysis and statistics - Please rephrase to Data and statistical analyses.
Answers for Q4-Q9: The reviewer did not care to point out the lines where these mistakes were noticed. It was a serious negligence of this reviewer. In spite of that, we have corrected some of these concerns. However, we do apologize to say that we do not agree to correct all of them.
Q10. Figure 1: The Negative cells title should be positioned a bit upwards. There also seems to be a broken line under HFD.
Answer: These are images obtained directly from the Flow cytometry machine, not drawn by the authors. Therefore, it is not possible to make any changes.
Q11. Figure 2 - The quality of the immunofluorescence photos is not the best.
Answer: We will improve the resolution of all the images as per the instructions of the IJMS Publishing Group, once the manuscript is accepted.
Q12. In the results section, general and literature data should not be included (such as: Aryl hydrocarbon receptor (AhR), a ligand-activated nuclear receptor and transcription
factor, has recently been implicated in mediating host-microbiota crosstalk).
Answer: What the reviewer is trying to say is not clear. What does “general and literature data” mean? We apologize that we will not be able to address this concern, as what the reviewer is trying to say is not clear.
Q13. 4.8. Immunostaining - Please correct. - Furthermore, this section contains the detailing of immunofluorescence, which is not the same as immunohistochemistry!
Answer: Corrected.
Round 2
Reviewer 1 Report
Comments and Suggestions for Authors
I have reviewed the revised manuscript and the authors’ responses to my initial comments. The authors have addressed all concerns satisfactorily . The manuscript has improved significantly and now meets the journal’s standards for publication. I recommend acceptance in its present form.